# Succinic Production from Source-Separated Kitchen Biowaste in a Biorefinery Concept: Focusing on Alternative Carbon Dioxide Source for Fermentation Processes

**Mariusz Kuglarz** [1,]* and **Irini Angelidaki** [2]

[1]  Faculty of Materials, Civil and Environmental Engineering, University of Bielsko-Biala, Willowa 2, 43-309 Bielsko-Biala, Poland
[2]  Department of Chemical and Biochemical Engineering, Technical University of Denmark, Soltofts Plads, 228 A, DK-2800 Kgs. Lyngby, Denmark
*   Correspondence: mkuglarz@ath.bielsko.pl

**Abstract:** This study presents sustainable succinic acid production from the organic fraction of household kitchen wastes, i.e., the organic fraction of household kitchen waste (OFHKW), pretreated with enzymatic hydrolysis (100% cocktail dosage: 62.5% Cellic® CTec2, 31%% β-Glucanase and 6.5% Cellic ® HTec2, cellulase activity of 12.5 FPU/g-glucan). For fermentation, *A. succinogenes* was used, which consumes $CO_2$ during the process. OFHKW at biomass loading > 20% (dry matter) resulted in a final concentration of fermentable sugars 81–85 g/L and can be treated as a promising feedstock for succinic production. Obtained results state that simultaneous addition of gaseous $CO_2$ and $MgCO_3$ (>20 g/dm$^3$) resulted in the highest sugar conversion (79–81%) and succinic yields (74–75%). Additionally, $CH_4$ content in biogas, used as a $CO_2$ source, increased by 21–22% and reached 91–92% vol. Liquid fraction of source-separated kitchen biowaste and the residue after succinic fermentation were successfully converted into biogas. Results obtained in this study clearly document the possibility of integrated valuable compounds (succinic acid) and energy (biogas) production from the organic fraction of household kitchen wastes (OFHKW).

**Keywords:** succinic acid; kitchen biowaste; organic fraction of household kitchen wastes; enzymatic hydrolysis; anaerobic digestion; carbon dioxide

## 1. Introduction

Bio-based production of succinic acid through microbial fermentation is currently considered as very attractive due to the fact that it can contribute to the abatement of $CO_2$ emissions, as this gas is consumed during succinic fermentation [1,2]. Among a number of bacterial strains able to generate succinic acid through anaerobic fermentation, *Actinobacillus succinogenes* is considered as one of the most promising for industrial applications, mainly due to its ability to ferment a wide range of carbon sources, i.e., glucose, xylose, arabinose, galactose, etc. [3,4]. Taking into account the fact that carbon dioxide is consumed during sugar conversion into succinic acid via *Actinobacillus succinogenes*, $CO_2$ supply is a crucial factor determining succinic yield, the ratio of succinic acid to other by-products and to the degree of sugar utilization [5–7].

Different types of biomasses have been tested as feedstock for microbial succinic production, which mostly include hydrolysed lignocellulosic biomass, e.g., hemp biomass, agricultural residues, wheat and rapeseed straw, as well as wastes (e.g., from food processing industry, e.g., rapeseed meal, citrus peel waste) and algae biomass [2]. Municipal biowastes (organic fraction of household kitchen waste, i.e., OFHKW) have not been widely used as feedstock for succinic production, especially with *Actinobacillus succinogenes*, which can be used with biogas as a carbon dioxide source. OFHKW, composed mainly of disposed food residues, is a rich source of degradable carbohydrates (from 30 to even 60–70% *w/w*)

and essential nutrients, including proteins (4–15% $w/w$), lipids (up to 10–15% $w/w$) and microelements (e.g., essential metals supporting bioconversion processes) [8,9]. It has been estimated that one-thirds to two-thirds of the annual food produced worldwide is not consumed and is disposed of as organic waste, with 88 million tons of food waste generated annually [10–12]. Source-separated municipal biowastes in the form of an organic fraction of household kitchen waste (OFHKW) can be a potential resource for the production of high-value compounds and biofuels. This includes the production of biogas, bioethanol, fatty acids, biopolymers, lactic acid and succinic acid [2,13–16]. Although the possibility of producing succinic acid (SA) from ruminal bacteria and food waste-related sources has been previously reported [2], integrating the SA production process with other value-added procedures such as biogas upgrading is appealing in terms of process effectiveness and economics. The proof of concept was previously studied by producing 14.4 g/L SA with a yield of 0.635 g/g of pure glucose together with a 35.4% ($v/v$) increase in methane content after 24 h of fermentation using *A. succinogenes* 130Z [4]. Later, through fermentation of hydrolysate after kitchen biowaste hydrolysis with *Basfia succiniciproducens,* succinic acid was produced in 0.46 and 0.25 g SA/g-glucose, using magnesium carbonate and raw biogas (including 40% of $CO_2$), respectively. Batch fermentation in the bioreactor with biogas resulted in 8.0% ($v/v$) of $CO_2$ decrease compared to raw biogas [8]. However, obtained effectiveness parameters [8] are significantly below the most optimal values obtained for feedstocks, such as mixed food waste, fruit and vegetables wastes, waste bread and bakery wastes (succinic yield: 0.67–1.18) [2].

This study presents sustainable succinic acid production from hydrolysed municipal biowastes (OFHKW), which are integrated with biogas production and purification. Succinic fermentation was carried out using *A. succinogenes* 130Z (ATCC 55618). The aim of this study was to analyse the influence of carbon dioxide (gaseous and solid $MgCO_3$) on the effectiveness of succinic fermentation. To the best of our knowledge, this is the first study evaluating the usage of a simultaneous $CO_2$ source (biogas after co-digestion processes and $MgCO_3$) for the production of succinic acid from municipal biowastes (OFHKW), with integrated biogas production from by-products after succinic fermentation.

## 2. Materials and Methods

### 2.1. Feedstock

The OFHKW sample used in this study originated directly from biowaste bins locally provided in the municipality of Bielsko-Biala (south of Poland). The OFHKW mainly consisted of food waste. The OFHKW sample used in this study originated from biowaste bins locally provided in the municipality of Bielsko-Biala (south of Poland, about 170,000 inhabitants). The OFHKW mainly consisted of food waste and was collected at a municipality level once per week from 20 households. Samples were collected between June and October (about 10 kg each, 20 OFHKW collections) from locations evenly distributed throughout the city of Bielsko-Biala. Once collected, the biowaste was mixed manually, and a representative sample (about 1 kg from each collection) was stored in a refrigerator at $-4$ °C. The sample used for enzymatic hydrolysis contained an even quantitative share of OFKKW from each collection (April–October, 20 individual OFHKW samples, based on weekly collections). The obtained sample was homogenized and autoclaved at 121 °C for 1 h, before chemical characterization (Table 1) and enzymatic hydrolysis (Table 2). OFHKW was separated into solid (85% of total sample weight) and liquid fractions (15% of total sample weight) (Table 1) using a low speed centrifuge (Benchtop, CFG-5BL, 2000 rpm). The solid fraction was enzymatically hydrolysed and used as feedstock for biosuccinic production, using *A. succinogenes* 130Z whilst the liquid fraction was used as feedstock for biogas production. Characteristics of the municipal biowastes were analysed using the methods described below, and they are presented in Table 1.

**Table 1.** Characterization of OFHKW (organic fraction of household kitchen waste) used in the current study (average values n = 3, ±standard deviation).

| Parameter | Unit | Value |
|---|---|---|
| Solid fraction of organic fraction of household kitchen waste ("OFHKW", 85% of total sample weight)[a] | | |
| Total solids (TS) | g/kg | 315 ÷ 31 |
| Volatile solids (VS) | g/kg | 293 ÷ 29 |
| Total organic carbon (TOC) | %TS | 57.3 ÷ 2.5 |
| Carbohydrates | %TS | 42.7 ÷ 3.1 |
| Cellulose | %TS | 29.2 ÷ 2.4 |
| Starch | %TS | 7.10 ÷ 0.4 |
| Hemicellulose | %TS | 6.40 ÷ 0.3 |
| Nitrogen (TKN) | %TS | 11.0 ÷ 0.55 |
| Lipid content | % TS | 8.46 ÷ 0.65 |
| Lignin | % TS | 8.90 ÷ 0.80 |
| Ash | % TS | 2.20 ÷ 0.15 |
| Ca | g/kg TS | 9.15 ± 0.80 |
| Mg | g/kg TS | 1.25 ± 0.10 |
| P | g/kg TS | 3.05 ± 0.12 |
| S | g/kg TS | 2.10 ± 0.11 |
| Na | g/kg TS | 3.55 ± 0.21 |
| K | g/kg TS | 4.52 ± 0.20 |
| Fe | g/kg TS | 0.52 ± 0.03 |
| Mn | mg/kg TS | <1.0 |
| Ni | mg/kg TS | 1.20 ± 0.10 |
| Cu | mg/kg TS | 0.82 ± 0.2 |
| Cd | mg/kg TS | <1.0 |
| Cr | mg/kg TS | <1.0 |
| Hg | mg/kg TS | <0.1 |
| Ni | mg/kg TS | <1.0 |
| Pb | mg/kg TS | <2.0 |
| Zn | mg/kg TS | 3.2 ± 0.4 |
| Liquid fraction of organic fraction of household kitchen waste ("OFHKW", 15% of total sample weight) [a] | | |
| pH | - | 4.8–5.2 |
| Total solids (TS) | g/dm$^3$ | 15.0 ± 1.2 |
| Volatile solids (VS) | g/dm$^3$ | 14.2 ± 1.2 |
| $NH_4^+$ | g/dm$^3$ | 0.62 ± 0.10 |
| $PO_4^{3-}$ | g/dm$^3$ | 0.11 ± 0.01 |
| VFA (acetic acid) | g/dm$^3$ | 0.55 ± 0.03 |
| Soluble sugars | g/dm$^3$ | 5.2 ± 0.6 |

[a]—average proportion of solid or liquid fraction in total sample weight, before grinding and enzymatic hydrolysis.

### 2.2. Enzymatic Hydrolysis

In order to break down the OFHKW complex into monomeric fermentable sugars for succinogenic bacteria, the solid fraction of OFHKW was hydrolysed with an application of β-Glucanase (from *Trichoderma longibrachiatum*—a mixture of enzymes with mainly β-1-3 and β-1-4-glucanase, xylanase, and cellulase activities, 40 FPU/g, G4423, Sigma-Aldrich) as well as Cellic® CTec2 (120 FPU/g) and Cellic® HTec2 (25 FPU/g) enzymatic cocktails. Enzymatic cocktails were purchased from Sigma-Aldrich® (Poznań, Poland). Cellulase activity, expressed as filter paper unit (FPU) per gram of enzyme solution, was determined by the Ghose method, established by the International Union of Pure and Applied Chemistry (IUPAC) [17].

Enzymatic dosage amounted to 5% *w/w*, calculated in relation to the substrate dry matter content, i.e., 5 g of the enzyme cocktails/100 g of substrate (OFHKW) dry matter. Percentage of individual enzymatic components (Cellic® CTec2, β-Glucanase and Cellic® Htec2) was calculated in relation to the total weight of the enzyme cocktail (100% enzymatic dosage represents, e.g., 5 g of the enzyme cocktails/100 g of OFHKW dry matter). The mixture of enzymatic cocktails used in this study (Table 2) contained: 62.5% Cellic® 103 Ctec2, 31%% β-Glucanase and 6.5% Cellic® Htec2, expressed as total weight of enzymatic cocktails. For reactions carried out with enzyme concentrations of up to about 4% (*w/w*) (preliminary results, data not shown), the amount of catalysts in the glucan and xylan hydrolysis is the only limiting factor, i.e., reaction rates are dependent only upon the level of enzymes. For higher doses of Cellic CTec2/β-glucosidase, rates of both glucose as well as xylose release were not directly proportional to the amount of enzyme present in reaction mixture (the rates versus enzyme concentration response become non-linear), most probably due to the saturation of substrate by enzymes [18]. Usage of enzyme concentrations above 5% (*w/w*) (>12.5 FPU/g-glucan) had no positive influence on the effectiveness of enzymatic hydrolysis. These enzymatic dosages correlated to between 0.3 and 1.25% of total sample volume (based results presented in Table 2) and cellulase activity of 12.5 FPU/g-glucan (10.5 FPU/g-carbohydrates or 4.5 FPU/g total biomass solids).

Hydrolysis was conducted at a solid loading of 6–25% in a 50 mM sodium citrate buffer, pH 5.4–5.5. Firstly, Erlenmeyer flasks (working volume of 100 cm$^3$) at 50 °C for 24 h were used. This part of the experiment was conducted in order to determine the most optimal biomass loading during enzymatic hydrolysis (Table 2). Secondly, kitchen biowaste hydrolysis was performed in 3 identical flasks of 10 dm$^3$ working volume at optimized conditions (wet substrate loading: 70% *w/v*, dry matter loading: 22%, Table 2). The aim of this part of the experiment was to prepare the required amount of hydrolysate for tests of succinic fermentation in bioreactors. Hydrolysates were filtered by sterile membrane filters (0.22 μm) and kept at 4 °C before further usage (fermentation tests).

### 2.3. Succinic Acid Fermentation in Bioreactors

Samples obtained after OFHKW hydrolysis, with biomass loading considered as the most optimal (Table 3), were used as feedstock for succinic acid production. Fermentation of hydrolysate was carried out in three identical 12 dm$^3$ bioreactors (Germany) with working volume of 10 dm$^3$. During start-up of the process, N$_2$ gas was used to create anaerobic conditions in fermenters.

Prior to start of batch fermentation, pH was adjusted to 6.8, with 50% H$_3$PO$_4$ and 0.05 mL of sterile Antifoam 204 added. Before addition of feedstock via sterile membrane filters, serum flasks with medium solution were autoclaved at 120 °C for 20 min. The fermentation was conducted for samples without synthetic medium (nutrients) as well as for hydrolysate mixed with nutrients. In each case, about 5% (*v/v*) of exponentially growing inoculum (OD$_{660}$ = 4.6–4.8) was added. The strain of *A. succinogenes* 130Z (DSM 22257) was obtained from DSMZ (German Collection of Microorganisms and Cell Cultures). Fermentation was conducted without minerals as well as with the following amounts of minerals: KH$_2$PO$_4$ (3 g/dm$^3$), MgCl$_2$ · 6H$_2$O (0.2 g/dm$^3$), CaCl$_2$ (0.2 g/dm$^3$),

NaCl ($1.0$ g/dm$^3$). In all fermentation assays, yeast extract ($10$ g/dm$^3$) was used as nitrogen source for fermentation processes [19].

In case of pH decrease below 6.8 due to acid production and insufficient carbonate–bicarbonate buffering capacity, NaOH solution (8 M) was added. Solid $MgCO_3$ (1.1 g $MgCO_3$/g-sugar), biogas (containing 75% of $CH_4$ and 25% of $CO_2$) as well as mixtures of magnesium carbonate and biogas were used as carbon sources (Table 3). The biogas originated from our previous sewage sludge and kitchen biowaste co-digestion experiments, conducted at lab scale in thermophilic conditions (52 °C). The biogas sample was collected during fermentation of kitchen biowaste (60% dry matter of the feedstock) and sewage sludge (40% dry matter of the feedstock), at optimized conditions (HRT = 25 days). The biogas originated from our previous sewage sludge and kitchen biowaste co-digestion experiments. This part of the experiment was conducted using 6:1 gas–liquid ratio and atmospheric pressure (101.3 kPa). The biogas was recirculated during fermentation, and changes in $CH_4$ and $CO_2$ concentrations were recorded. In addition, 1 mL samples were taken periodically (after 0, 3, 6, 12, 15, 18, 24, 36 and 48 h) and used for analysis of sugars (glucose, xylose) and acids (succinic-, acetic- and formic). Succinic acid yield ($Y_{SA}$) was calculated as the amount of succinic acid (g/dm$^3$) obtained per 1 g/dm$^3$ of sugars consumed. Sugar utilization was calculated as the difference between initial sugar content (g/dm$^3$) and sugar content after succinic acid production (g/dm$^3$).

### 2.4. Anaerobic Digestion (AD)

Liquid fractions, originating from "OFHKW" as well as solid residues after enzymatic hydrolysis (post-hydrolysis residues), were tested as feedstock for methane production. Biochemical methane potential (BMP) tests were determined in batch experiments, in triplicates. The experiments were performed in 540 cm$^3$ serum glass bottles with working volume of 250 mL, at 2 g VS/dm$^3$.

Digestate from a full-scale plant (52 °C) treating manure and kitchen waste was used as inoculum. Methane produced from inoculum was subtracted from the assays. Bottles were flushed with pure $N_2$ for 3–5 min, sealed with rubber stoppers and aluminium crimps and finally incubated at 52 °C until no significant amounts of $CH_4$ were produced (approx. 30 days). Methane yields are expressed as m$^3$ $CH_4$/kg VS$_{added}$ at standard temperature and pressure (0 °C, 100 kPa).

### 2.5. Calculations

2.5.1. Enzymatic Hydrolysis

Glucan and xylan yields during enzymatic hydrolysis were calculated according to Formulas (1) and (2):

$$\text{Glucan}_{\text{Yield}} \ (\%) = \frac{\text{Glucose}_{\text{Released}}}{\text{Glucan}_{\text{Feedstock}} \cdot \left(\frac{180}{162}\right)} \cdot 100 \tag{1}$$

$$\text{Xylan}_{\text{Yield}} \ (\%) = \frac{\text{Xylose}_{\text{Released}}}{\text{Hemicellulose}_{\text{Feedstock}} \cdot \left(\frac{150}{132}\right)} \cdot 100 \tag{2}$$

where:

$\text{Glucose}_{\text{Released}}$ and $\text{Xylose}_{\text{Released}}$—the amount of glucose and xylose released during enzymatic hydrolysis, g;

$\text{Glucan}_{\text{Feedstock}}$—total amount of starch and cellulose-derived glucan in organic fraction of household kitchen waste, g;

$\text{Hemicellulose}_{\text{Feedstock}}$—total amount of hemicellulose compounds in organic fraction of household kitchen waste, respectively, g;

180/162 and 150/132—stoichiometric conversion factors of glucan to glucose and hemicellulose to xylose, respectively.

### 2.5.2. Succinic Fermentation

Succinic acid yield ($Y_{SA}$) was calculated as the amount of succinic acid (g) obtained per 1 g of sugars (glucose + xylose) consumed (Equation (3)).

$$Y_{SA}(\%) = \frac{SA_{Prod.}}{Sugar_{Consumed}} \cdot 100 \qquad (3)$$

where:

$SA_{Prod.}$—concentration of succinic acid produced ($g/dm^3$);

$Sugar_{Consumed}$—amount of glucose and xylose consumed during succinic acid fermentation ($g/dm^3$).

### 2.6. Analytical Methods

Total solids (TS), volatile solids (VS), and ash and nitrogen content were determined according to standards methods [20]. pH was measured using a standard pH meter (Aldrich® glass pH electrode, Z113077-1EA). The content of cellulose, hemicellulose and lignin in raw material as well as solid residues after enzymatic hydrolysis were determined according to the National Renewable Energy Laboratory (NREL) analytical methods for biomass characterization [21]. Concentrations of sugars and organic acids (succinic, acetic, formic) were measured by using high performance liquid chromatography HPLC (Agilent 1260 Infinity, Germany) equipped with a BioRad Aminex HPX-87H column at 63 °C and ultraviolet (UV) and refractive index (RI) detector (67162A, Germany), using 4 mM $H_2SO_4$ as eluent at 0.6 mL·min$^{-1}$ flow rate. To protect the HPX-87H column from contamination and foreign particles, a guard column was fitted to the system. Concentrations of (Ca, Mg, Na, S) as well as metals were analysed in a Vista-MPX inductively coupled plasma spectrometer with optical emission spectrometry, ICP-OES (Agilent, Santa Clara, CA, USA). Concentration of phosphates and ammonia–nitrogen was analysed by spectrophotometric method, using standard cuvette tests (HACH-LANGE). Lipid content was analysed after extraction using the Soxhlet method. VFAs were measured by gas chromatography (GC) with flame ionization detection (FID). Total organic carbon (TOC) was determined by titration method with potassium dichromate, after sample mineralization with sulfuric acid. Content of $CH_4$ and $CO_2$ in biogas before and after upgrading was analysed with application of gas chromatography (GC), Thermo Scientific with a TCD-detector and an HP-plot column.

## 3. Results and Discussion

### 3.1. Characterization of the Organic Fraction of Household Kitchen Waste (OFHKW)

The characterization of OFHKW used in this study is presented in Table 1. High organic content (VS/TS ratio of 93%, C5 and C6 sugars: 40–46% of TS) proves that OFHKW (organic fraction of household kitchen waste) can be treated as a promising substrate for succinogenic bacteria. Glucan content, which represents cellulose and starch, accounted for about 85% of all carbohydrates, whilst hemicellulose fraction was on average 15% of total carbohydrates (based on results in Table 1). This is in agreement with results obtained by [22]. This substrate also contains a high content of proteins, i.e., up to 10–12% of total solids, and is in the range found in food and bakery wastes [8].

It was also found that kitchen biowaste (OFHKW) contains an insignificant amount of heavy metals, whilst containing essential elements for microbial growth, including: magnesium, calcium, cobalt, and nickel (Table 1).

### 3.2. Enzymatic Hydrolysis of Solid Fraction from OFHKW

In all enzymatic tests, enzyme contents did not exceed 1.5% of total weight sample, which corresponds to 12.5 FPU/g-glucan (10.5 FPU/g-carbohydrates or 4.5 FPU/g dried substrate, based on results in Table 2). Therefore, applied enzyme dosages can be considered as relatively low, compared to other biomass types, hydrolysed at high biomass loadings.

For example, pretreated sugarcane bagasse by the Organosolv method was hydrolysed at 10 FPU/g dried substrate. In general, due to the recent scientific progress in the field of enzymatic hydrolysis, the acceptable cellulase dosage used for biomass hydrolysis is around or even below 10 FPU/g dry matter [23].

The effectiveness of the glucan conversion reached 93% for the wet substrate loading between 20 and 40% (dry matter loading 6–13%) (Table 2, Figure 1). This can be explained by relatively high activity of carbohydrate-degrading enzymes present in this enzyme blend used. Slightly lower glucan conversion yields (79–84%) were obtained for biomass wet loading in the range of 60–70% (18–22% dry matter loading). However, lower conversion yields were compensated for by higher glucose concentrations. i.e., 63–70 g/dm$^3$. For assays with biomass loading not exceeding 22% (*w*/*v*, dry matter), glucan and xylan were hydrolysed without a log phase (Figure 1A,B). In the case of the sample with the highest tested biomass loading (dry matter loading: 25.0%, Figure 1C), release of glucose and xylose started after about 6 h of lag phase, which shows that process conditions were not sufficiently optimal. Moreover, enzymatic processes conducted for the biomass wet loading of 80% (dry matter loading: 25%) was connected with significantly lower conversion yields (69%) compared to assays with lower biomass loadings (Table 2). In this case, an average obtained glucose concentration (74 g/dm$^3$) is not beneficial, taking into account a higher required dosage of enzymatic cocktails used for processing an assay with 80% biomass wet loading. A significant decrease in glucan hydrolysis was observed after increasing dry matter loading from 22% to 25% (Table 2). Taking into account the fact that the difference between analysed assays is relatively low, this phenomenon might have been caused by reduced viscosity or reduction in free water availability.

**Table 2.** Results of enzymatic hydrolysis carried out with selected cocktails mixture (average values n = 3, ± standard deviation) [a].

| Assay | Initial Glucose and Xylose [b], g/dm$^3$ | After 24 h of Enzymatic Hydrolysis | | | | | |
|---|---|---|---|---|---|---|---|
| | | Glucose, g/dm$^3$ | Glucan Yield, % | Xylose, g/dm$^3$ | Xylan Yield, % | Total Sugar, g/dm$^3$ | Total Yield, % |
| Wet substrate loading: 20% (% *w*/*v*) Dry matter loading: 6.3% ± 0.6 | 29.7 ± 0.9 | 23.6 ± 1.1 | 93.3 ± 1.5 | 3.75 ± 0.2 | 82.3 ± 2.5 | 27.3 ± 1.2 | 91.7 ± 1.1 |
| Wet substrate loading: 40% (% *w*/*v*) Dry matter loading: 12.5% ± 1.2 | 59.6 ± 1.9 | 47.0 ± 3.0 | 93.0 ± 3.6 | 7.40 ± 0.5 | 81.0 ± 2.6 | 54.4 ± 3.4 | 91.2 ± 3.3 |
| Wet substrate loading: 60% (% *w*/*v*) Dry matter loading: 18.9% ± 1.9 | 89.4 ± 2.9 | 63.4 ± 3.8 | 83.6 ± 2.5 | 9.75 ± 0.5 | 71.3 ± 1.5 | 73.2 ± 4.2 | 81.8 ± 2.3 |
| Wet substrate loading: 70% (% *w*/*v*) Dry matter loading: 22.0% ± 2.2 | 104 ± 3.4 | 70.0 ± 3.1 | 79.3 ± 3.0 | 10.8 ± 0.7 | 67.6 ± 2.5 | 80.8 ± 3.6 | 77.5 ± 2.7 |
| Wet substrate loading: 80% (% *w*/*v*) Dry matter loading: 25.0% ± 2.5 | 119 ± 3.8 | 73.9 ± 6.1 | 69.0 ± 3.6 | 11.1 ± 0.9 | 61.0 ± 3.6 | 85.0 ± 6.8 | 71.3 ± 3.4 |

[a]—enzymatic cocktail weight expressed as substrate dry matter used during experiment (5% *w*/*w*), 100% dosage: 62.5% Cellic® CTec2, 31%% β-Glucanase and 6.5% Cellic ® HTec2, [b]—theoretical (maximum) glucose and xylose concentration, after enzymatic hydrolysis and mixing with exponentially growing inoculum (5% *v*/*v*).

In the case of xylan hydrolysis, obtained yields varied between 61% and 81% (Table 2). Obtained xylan yields can be considered as very high, due to the enrichment of enzymatic dosage with the Cellic ® HTec2 cocktail and β-Glucanase. This was also observed in previous studies [18]. Similarly to cellulose hydrolysis, conducting the process at the highest biomass loading (dry matter loading: 25.0%, Figure 1C) had no positive effect on the final xylose concentrations and xylose yields (Table 2). Final xylose concentrations were several times lower compared to glucose titres, as carbohydrates contained only 15% of hemicellulosic fraction. In all cases, there was no significant influence of process duration (above 24 h) on the final glucose/xylose concentrations or glucan/xylan yields. In this study, final sugar content amounted to 80.8 g/dm$^3$ (Table 2, enzymatic assay considered as optimal, substrate loading: 70% *w/v*, dry matter loading: 22.0%), which can be considered as a promising feedstock for biosuccinic acid production. This is within an upper range of values reported after enzymatic hydrolysis of kitchen biowastes and other organic wastes, originating from the food industry. For example, treatment of bakery wastes and food wastes (both after fungal autolysis and hydrolysis) resulted in total sugar titres of 54.2 and 31.9 g/dm$^3$, respectively [24,25], whilst treatment of herbal extraction residues (after dilute-acid pretreatment before enzymatic hydrolysis) obtain 71.6 g/dm$^3$ released sugars [26]. Higher concentrations of sugars (119–135 g/dm$^3$) were obtained by Babaei et al. [8]. However, in this case, feedstock (organic fraction of kitchen waste, OFHKW) which was directed to enzymatic hydrolysis contained three times higher content of total solids (TS) and volatile solids (VS) (TS: 976 g/kg, VS: 921 g/kg, Celluclast 1.5 L®, Cellobiase, substrate loading: 15–20%), compared to the current study. Taking into account the above, hydrolysed feedstock in our studies (>80 g/dm$^3$, Table 2) can be considered as promising for biosuccinic acid production. Our results also indicate that the strategy of mixing commercially available cellulolytic enzyme preparations may in the future contribute to the development of improved enzymatic mixtures for municipal biowaste hydrolysis.

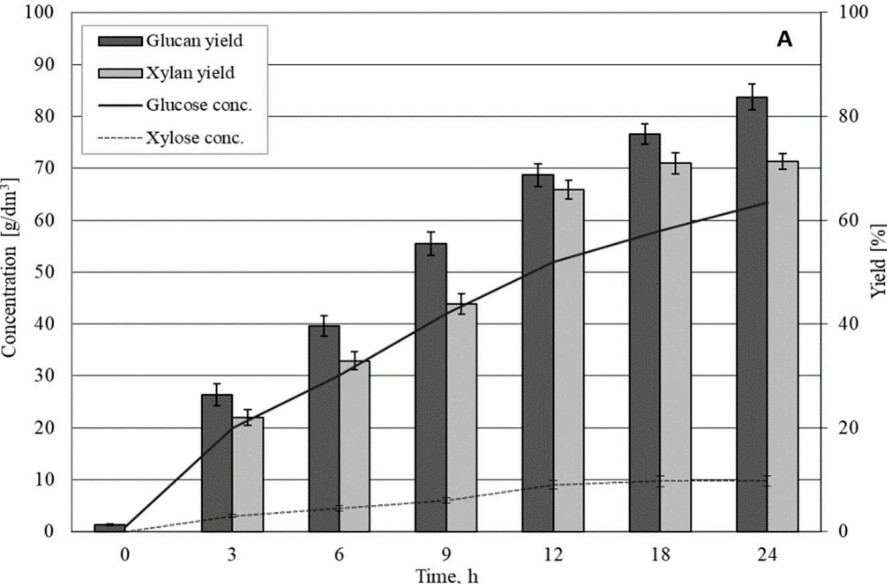

**Figure 1.** *Cont.*

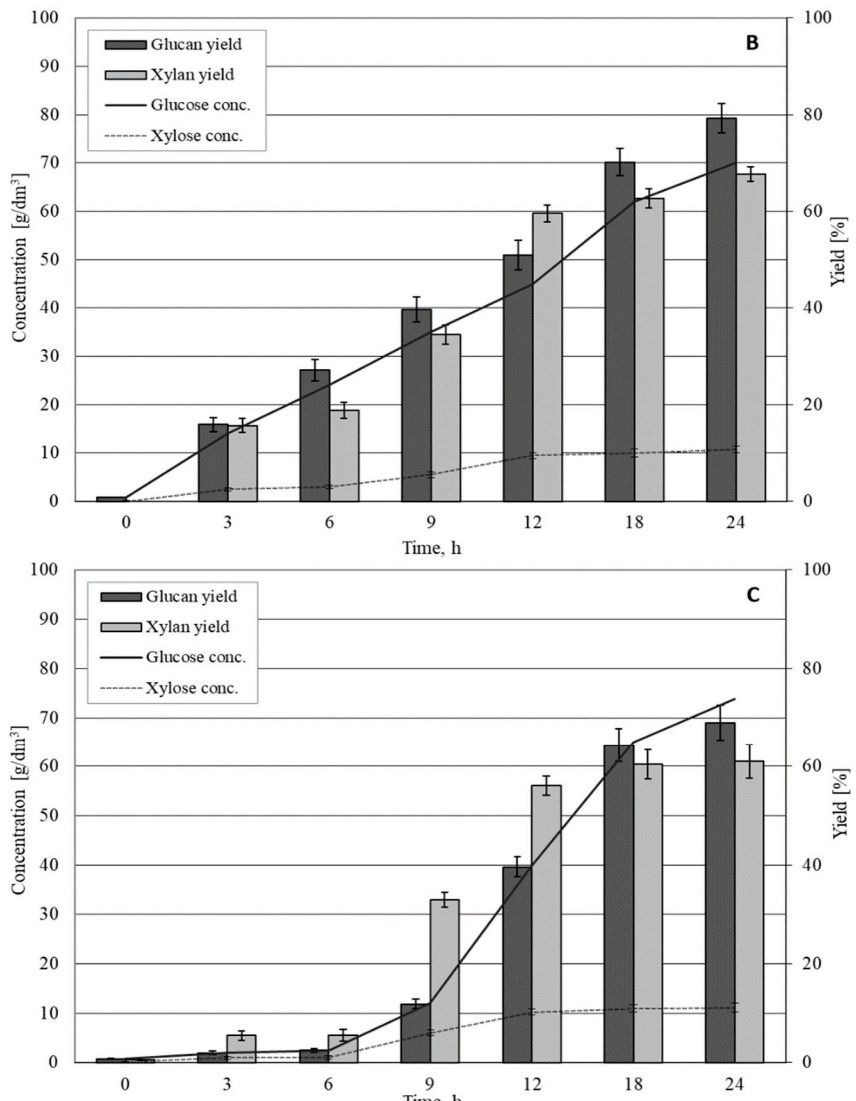

**Figure 1.** Selected course of enzymatic hydrolysis and obtained glucose and xylose yields ((**A**)—18.9% dry matter loading/wet substrate loading: 60% *w/v*; (**B**)—22% dry matter loading/wet substrate loading: 70% *w/v*; (**C**)—25% dry matter loading/wet substrate loading: 80% *w/v*; error bars represent standard deviation).

### 3.3. Succinic Fermentation Using Different Carbon Sources

Carbon dioxide constitutes one of the major substrates used for biosynthesis of succinic acid, while $CO_2$ source and dosage influence the metabolic flux as well as the effectiveness of succinic acid production [27,28]. Firstly, succinic acid fermentation was conducted using gaseous $CO_2$ (biogas containing 75% $CH_4$ and 25% $CH_4$). The process conducted at atmospheric pressure (101.3 kPa, $CO_2$ partial pressure 40 kPa) helped to utilize about 42% of available sugars (glucose and xylose) and resulted in succinic yield of 63% (0.63 g succinic acid/g sugar consumed). Succinic yields obtained were slightly lower than yields previously reported, using model biogas mixture (60% $CH_4$ and 40% $CO_2$) as the carbon source for succinic acid biosynthesis [4]. However, the present study is based on biogas containing significantly lower $CO_2$ content (25% vol.) compared to previous studies (Table 3). Taking into account the conditions applied in the current study, it was assumed that providing biogas after co-digestion processes (including 25% of $CO_2$) as the only carbon dioxide source is not sufficient for obtaining a high succinic titre (above 30–40 g/$dm^3$) and succinic yields > 65–70% (Table 3). Obtaining a high succinic titre (at least 30–40 g/$dm^3$) during fermentation is crucial, as downstream processing of succinic

broths can utilize more than 50–60% of total production costs and is attributed to recovery and refining [2]. Conditions of biogas supplied as the only carbon dioxide source can be further optimized (e.g., via changing gas–liquid ratio); however, this is outside the scope of the current study.

**Table 3.** Succinic acid production from hydrolysates after application of the most effective enzyme dosage (initial sugar concentration: $76.8 \pm 3.5$, average values n =3, $\pm$ represent standard deviations).

| CO$_2$ Source | Nutrients for Ferm. [a] | After Succinic Fermentation (48 h) | | | | | Biogas after Succinic Production [g] | |
|---|---|---|---|---|---|---|---|---|
| | | Glucose, g/dm$^3$ | Xylose, g/dm$^3$ | Sugar Utiliz., % | Succinic Acid, g/dm$^3$ | Succinic Yield (Equation (3)), % [f] | CH$_4$ % vol. | CO$_2$ % vol. |
| MgCO$_3$ (85–86 g/dm$^3$) [b] | + | $13.8 \pm 1.9$ | $2.79 \pm 0.3$ | $78.5 \pm 3.2$ | $43.7 \pm 3.0$ | $72.4 \pm 2.8$ | - | - |
| MgCO$_3$ (85–86 g/dm$^3$) [b] | - | $14.1 \pm 1.3$ | $2.62 \pm 0.4$ | $78.1 \pm 1.8$ | $42.3 \pm 2.5$ | $70.5 \pm 2.4$ | - | - |
| Biogas as CO$_2$ source | + | $36.9 \pm 3.5$ | $6.67 \pm 0.8$ | $42.4 \pm 3.4$ | $21.7 \pm 1.7$ | $63.0 \pm 4.4$ | $83.5 \pm 1.6$ | $17.2 \pm 1.0$ |
| MgCO$_3$ (30–31 g/dm$^3$) [c] + Biogas | - | $11.9 \pm 2.2$ | $2.40 \pm 0.5$ | $81.1 \pm 3.6$ | $46.3 \pm 2.0$ | $74.9 \pm 3.3$ | $87.5 \pm 1.4$ | $11.1 \pm 1.0$ |
| MgCO$_3$ (20–21 g/dm$^3$) [d] + Biogas | - | $13.2 \pm 1.0$ | $2.84 \pm 0.3$ | $79.0 \pm 1.5$ | $45.7 \pm 2.5$ | $75.4 \pm 4.8$ | $91.2 \pm 1.5$ | $8.65 \pm 0.9$ |
| MgCO$_3$ (14–15 g/dm$^3$) [e] + Biogas | - | $24.5 \pm 2.1$ | $2.92 \pm 0.5$ | $64.4 \pm 2.7$ | $34.3 \pm 1.2$ | $69.5 \pm 2.1$ | $90.2 \pm 1.4$ | $9.25 \pm 0.8$ |

[a]—"+"—nutrients (KH$_2$PO$_4$: 3 g/dm$^3$, MgCl$_2$. 6H$_2$O: 0.2 g/dm$^3$, CaCl$_2$: 0.2 g/dm$^3$: NaCl (1.0 g/dm$^3$) and yeast extract were added, "-"—fermentation with yeast extract/without nutrients; [b]—1.1 g MgCO$_3$/g-sugar considered as theoretical dosage; [c]—35% of theoretical MgCO$_3$ dosage; [d]—25% of theoretical MgCO$_3$ dosage; [e]—15% of theoretical MgCO$_3$ dosage; [f]—calculated as g succinic acid produced/g sugar consumed according to Equation (3), [g]—initial composition of biogas (CH$_4$: 75% vol, CO$_2$: 24% vol.).

In many previous studies on biosuccinic acid production, salts (MgCO$_3$, NaHCO$_3$ or CaCO$_3$) are used as the carbon source. In particular, MgCO$_3$ as a carbon source was identified as the most effective CO$_2$ supplier and pH control agent [2]. During the second part of the experiment, MgCO$_3$ was used as a carbon dioxide source for succinic production. In these cases, succinic titre and yield amounted to 42–44 g/dm$^3$ and 70–72%, respectively (Table 3). This proves that the availability of CO$_2$ plays a crucial role during succinic acid biosynthesis and promotes the carbon flow towards the SA production branch of TCA (tricarboxylic acid) cycle [1,2]. Obtained succinic yields and titres were similar in both analysed cases, i.e., with the addition of nutrients for fermentation and for the process without nutrient addition (Table 3). This proves that the organic fraction of household kitchen waste contains essential elements for microbial growth, including: magnesium, calcium, cobalt, nickel (Table 1).

As our main aim was to utilize CO$_2$ from biogas for effective succinic production, the simultaneous effect of gaseous CO$_2$ and MgCO$_3$ on the performance of succinic fermentation was taken into account (Table 3). When MgCO$_3$ was added with the supply of gaseous CO$_2$, the following compounds, CO$_2$, HCO$_3^-$ and CO$_3^{2-}$, would become equilibrised in the fermentation broth [5]. The highest succinic titre and yield amounted to 45–46 g/dm$^3$ and 74–75%, respectively, and these values were obtained for a MgCO$_3$ dosage of 20–30 g/dm$^3$ MgCO$_3$+ biogas/(>400 mM CO$_2$) (Table 3). Succinic acid production started immediately after inoculation, without any lag phase (Figure 2). In these conditions, the dosage of carbon sources significantly exceeded the maximum CO$_2$ solubility (139 mM). This proves that high succinic yields are obtained in conditions of CO$_2$ excess. This is in accordance with previous reports stating that the highest succinic yields (69–71%, 0.69–0.71 g/g) were recorded for carbonate dosages of 300–500 mM [7]. This also proves that biogas can substitute the majority of MgCO$_3$, which is commonly used as a carbon source in biotechnological processes.

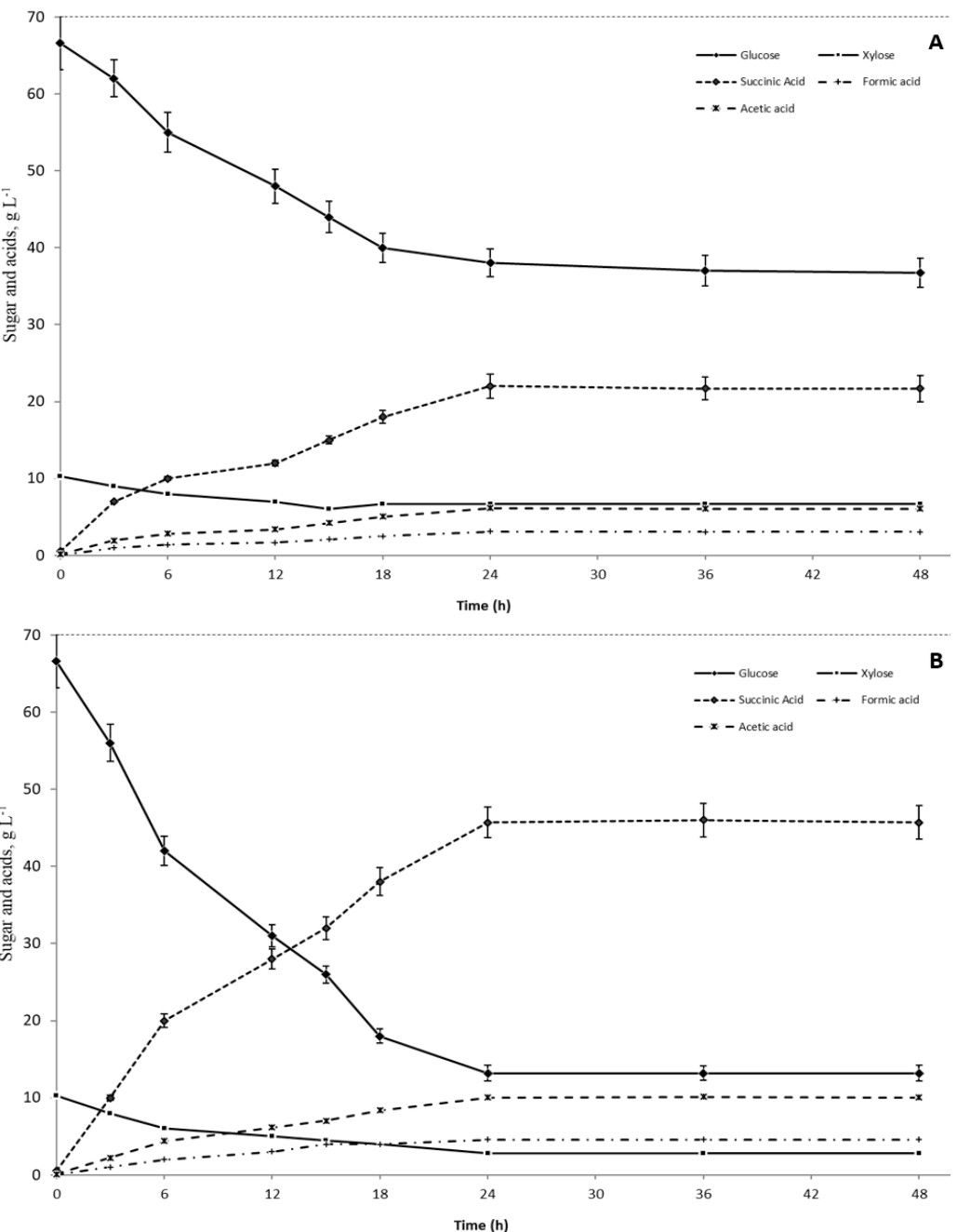

**Figure 2.** Course of succinic acid fermentation ((**A**)—process carried out with biogas as $CO_2$ source, (**B**)—process carried out with simultaneous addition of biogas and 20–21 $g/dm^3$ $MgCO_3$/25% of initial dosage; error bars represent standard deviation).

An additional purpose of using gaseous $CO_2$ for the biosynthesis of succinic acid was to purify the biogas. $CH_4$ content in biogas used as the only $CO_2$ source increased by 8–9% and reached the level of 83–84% at the end of succinic fermentation (Table 3). An increase in $CH_4$ content from 85% to 95% was reported in previous studies [4]. However, as previously mentioned, the authors used biogas containing significantly higher $CO_2$ content (40% vol.) compared to the present study (25% vol.).

As regards the simultaneous usage of biogas and $MgCO_3$ as carbon dioxide sources (dosage of 15–20 $g/dm^3$, 15–25% of theoretical $MgCO_3$ dosage, Table 3), $CH_4$ content after succinic fermentation reached final values of 90–91% (Table 3). In these cases, the final $CH_4$ content is consistent with values that are possible to obtain via commercially

available purification methods, e.g., water scrubbing or using chemical scrubbers with amine solutions [29]. In the case of simultaneous addition of biogas and 30–31 g/dm$^3$ $MgCO_3$ (35% of theoretical $MgCO_3$ dosage, Table 3), the $CH_4$ content in biogas fluctuated between 87 and 88%. However, such process conditions reflected a high $CO_2$ content, originating from carbonate, compared to previous assays. Depending on succinic fermentation conditions, including type of $CO_2$ source and its dosage, other metabolites, such as acetic, formic, lactic acid or ethanol, can be produced in different amounts [1]. In the present study, acetic and formic acids were produced as the main fermentation by-products. As shown in Figure 2, during fermentation with the simultaneous addition of biogas and $MgCO_3$, succinic, acetic and formic acids were simultaneously produced, and by-product content (acetic and formic acid) did not exceed 24–25% of total fermentation (Figure 3). This shows that applied $CO_2$ dosage was sufficient to increase C4 flux towards higher succinic acid production [1]. Only in the case of fermentation with biogas as the only $CO_2$ source did by-product content slightly exceed 30% of total fermentation products (succinic, acetic and formic acid) (Figure 3).

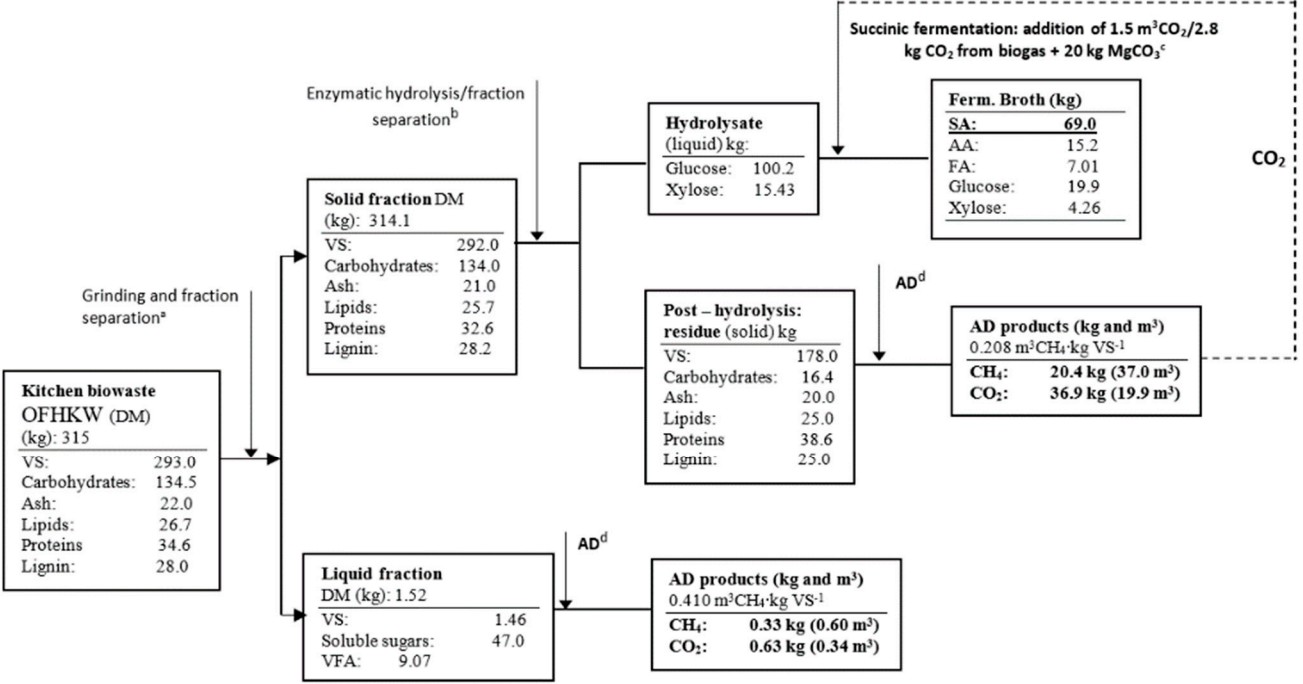

**Figure 3.** Simplified mass balance of OFHKW complex (1 mg of waste treated), maximizing the products in a biorefinery approach (a—mechanical grinding and fraction separation via centrifugation, b—enzymatic hydrolysis conducted at wet substrate loading: 70% dry matter loading: 22.0, enzymatic cocktail weight expressed as substrate dry matter used during experiment: 5% *w/w*), c—succinic fermentation with 25% of theoretical $MgCO_3$ dosage, simultaneous addition of biogas and $MgCO_3$: 20–21 g/L, d—anaerobic digestion of post.

### 3.4. Anaerobic Digestion (AD) of Succinic By-Products

In the case of liquid fraction of the OFHKW complex, obtained methane yields amounted to 0.410 ± 25 Nm$^3$ $CH_4$/kg $VS_{added}$ (Figure 3). Significantly lower methane yields were obtained after treatment of post-hydrolysis residues. In this case, methane yields fluctuated between 0.200 and 0.220 Nm$^3$ $CH_4$/kg $VS_{added}$ (208 ± 11Nm$^3$ $CH_4$/kg $VS_{added}$) (Figure 3). This is within the range of methane yields reported for AD of kitchen waste, i.e., 0.175 and 0.240 Nm$^3$ $CH_4$/kg $VS_{added}$ [30]. Higher methane yields obtained in the case of liquid OFHKW fraction were accounted for in the presence of easily biodegradable compounds (i.e., acetic acid, dissolved sugars, lipids). This in consistent with the fact that rapid bioconversion of dissolved components is usually observed. This can also be

accounted for in high biodegradability of the liquid fraction ("OFHKW"), which was above 0.92 (based on COD and $BOD_5$ results, data not shown). In all analysed cases, the $CH_4$ and $CO_2$ content in biogas produced amounted to 64–66% and 32–34%, respectively (Figure 3).

*3.5. Biorefinery Concept*

In the case of succinic acid production from the solid fraction of OFHKW, 69 kg of succinic acid/Mg of treated feedstock was obtained (Figure 3). Produced succinic acid has been recognized as one of the twelve most promising building-block chemicals, and it is a precursor for the production of a wide spectrum of commodities used in the food, chemical, and pharmaceutical industries. In our previous studies, succinic acid from biomass was successfully purified via the integrated concept of cation exchange (Amberlite IR 120H, cation exchange resin, vacuum distillation and crystallization). The developed concept of broth purification helped to obtain succinic acid with high quality (>98%), which can be used for further chemical transformations [18]. In the present biorefinery approach, post-hydrolysis residues (solid fraction after enzymatic treatment) and liquid fraction of "OFHKW" were used as feedstock for biogas production, i.e., a process which generates valuable methane and carbon dioxide (Figure 3). In the lab conditions presented in this study, carbon dioxide production in the form of biogas exceeds the dosage of $CO_2$ used for effective succinic production (Table 3, Figure 3). Therefore, it is evident that the biorefinery concept presented has a real chance of contributing to the latest trends connected with the abatement of $CO_2$ emissions from biofuel/biochemical production. Moreover, excessive amounts of $CO_2$ can be used in numerous industrial applications, e.g., for soft drink and soda water production. It is estimated that 1 tonne of biosuccinic acid produced can save 4.5–5 tonnes of $CO_2$ compared to succinic production via petrochemical sources. Furthermore, the effectiveness of biosuccinic acid production from analysed biowaste can be improved via the optimization of the fermentative process, including *Actinobacillus succinogenes* adaptation into process conditions [31].

**4. Conclusions**

The results obtained in this study clearly confirmed that OFHKW after enzymatic hydrolysis (mixing different enzymatic cocktails: Cellic® CTec2, β-Glucanase and Cellic® HTec2) can be considered as a promising feedstock for succinic acid production. Enzymatic hydrolysis of kitchen waste at high biomass loading (>20%) resulted in a total sugar yield of 78%. Succinic fermentation with the simultaneous addition of gaseous $CO_2$ (biogas) and $MgCO_3$ (>20 g/dm³) resulted in the highest sugar conversion rates (79–81%) and succinic yield (74–75%). The concept of kitchen waste treatment presented in the study turned out to be effective in biogas production from residues after succinic production, especially for post-hydrolysis biomass. Optimal conditions of succinic fermentation ($CO_2$ dosage and source) identified in this study can pave the way towards the sustainable production of succinic acid from the organic fraction of municipal wastes (OFHKW), using a biorefinery concept.

**Author Contributions:** M.K. performed experimental research during his scientific stay at DTU and via cooperation with their home University of Bielsko-Biala. M.K. also prepared a draft version of the manuscript. I.A. supervised the research performed and final preparation of the manuscript. All authors have read and agreed to the published version of the manuscript.

**Funding:** This research was supported and financed by NAWA (National Agency for Academic Exchange), via project "Production of succinic acid from municipal bio-fraction based on waste carbon dioxide and alternative nitrogen sources" (PPN/BEK/2019/1/00411) and ACT ERA-NET Cofund under the European Union's Horizon 2020 Research and Innovation program (Project No. 327331 CooCE, Grant Agreement 862087), co-funded by EUDP (No 64021.2006).

**Institutional Review Board Statement:** Not applicable.

**Informed Consent Statement:** Not applicable.

**Data Availability Statement:** Not applicable.

**Acknowledgments:** The research conducted was supported and financed by NAWA (National Agency for Academic Exchange), via project "Production of succinic acid from municipal bio-fraction based on waste carbon dioxide and alternative nitrogen sources" (PPN/BEK/2019/1/00411). Experimental results were obtained during a scientific stay at DTU (Technical University of Denmark, 2020–2021). Moreover, it was supported by the ACT ERA-NET Cofund under the European Union's Horizon 2020 Research and Innovation program (Project No. 327331 CooCE, Grant Agreement 862087), co-funded by EUDP (No 64021.2006).

**Conflicts of Interest:** The authors declare no conflict of interest.

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
