# Peer review of "Succinic Production from Source-Separated Kitchen Biowaste in a Biorefinery Concept: Focusing on Alternative Carbon Dioxide Source for Fermentation Processes"

_fermentation, doi:10.3390/fermentation9030259_

Round 1
Reviewer 1 Report
This study presents a strategy for succinic acid production from the organic fraction of municipal wastes having a biorefinery approach. The subject is highly interesting for researchers in the field and is within this journal's scope. The experimental strategy is also appropriate. I recommend the publication of this study after some modifications I point out in detail below. Mainly, I think some parts could be better described (especially in the methods part), and deeper discussion should appear in the "results and discussion" section.
Line 51. It is better to define the concept of "source-separated municipal biowastes" clearly here, as this may not be obvious for people in countries where this practice is not common. Also, revise in the text the term "source-separated" that sometimes appears as "source separated". It would also be interesting to add some comments and data on the availability and logistics for using this material in a biorefinery in countries where source-separated municipal biowastes are collected.
OFHKW needs to be defined only once in the text. After that, make consistent use of the acronym created. (except in tables and Figures, where it is best to define it again)
Section 2.1. How the sampling of OFHKW occurred? Did the authors get samples from different bins? How would changes in composition change the results obtained in the best conditions? Have you tested different samples of OFHKW to be sure that the process is robust enough, even with variations in the raw material composition? This is an essential factor to evaluate when proposing this type of raw material. Please, comment on that. It is better to describe the methods for characterizing the biomass here; otherwise, we see much-related information, but the characterization methods are listed only in the last topic. Also, please describe de glucan and xylan content (or the whole composition determined by the NREL protocol), as this is not shown in Table 1, and Table 2 presents results for glucose and xylose yields.
Section 2.2. The term "second-generation enzymes" may not be familiar to many readers, as this is a denomination in the context of enzyme development for cellulose hydrolysis on cellulosic ethanol production. It is also better to specify the activities of the enzymes and their sources in another way. Describing the FPU units in the middle of the description is confusing. From where Cellic enzymes were obtained? Were they purchased from Sigma or donated by Novozymes? Specify. Add the reference for Ghose method. The enzyme dosage used (5% w/w) is unclear. As I understood, this 5% is in relation to the substrate dry matter content, i.e., if 100 g of substrate were to be used, then we need to add 5g of the enzyme cocktail. Is that so? I had to read the text more than once and combine it with the table information to reach this conclusion, so I guess it could be better described in the text. Also, the preparation of the enzyme cocktail could be better described. "100% cocktail dosage: 62.5% Cellic® 103 CTec2, 31%% β-Glucanase and 6.5% Cellic ® HTec2, expressed as a percentage of substrate 104 dry matter" is misleading. I guess the percentages are in relation to the total weight of the enzyme cocktail and not the substrate dry matter, right?
Section 2.5.1. Formula [1] – it is better to state "glucan" and not "cellulose" as glucose in hydrolysates may be derived from starch in the OFHKW and NREL protocol does not discriminate between starch and cellulose-derived glucan, unless previous hydrolysis with amylases is performed to determine the starch content separately. Did the authors determine the starch content in the residue? Considering that the biomass comes from kitchen waste, I expect a considerable amount of starch, which influences more efficient hydrolysis at a high dry matter than a pure-cellulosic source of glucose. Indeed, in lines 216-217 authors state, "Glucan content, which represents cellulose and starch, accounted for about 85% of all carbohydrates."
Table 2. What is the initial glucose and xylose content? I do not understand. Is that the theoretical maximum?
Section 3.1. As commented above, I think it is very important to comment on how this material composition changes when different sampling is done and how this impacts a biorefinery or facility based on the use of this material. How representative is the compositional data presented in this paper if one decides to use the biowaste from Bielsko-Biala, for example? Collecting samples at a different seasons would result in a very different composition? Please, include thoughts on that.
Section 3.2. Much information in here repeats what is already stated in the materials and methods. I recommend better describing the assays in the M&M, so everything is clear. Then, here, the authors can focus on discussing the results.
I am interested to know which of those enzymes contain amylases since the biomass used contains starch in its composition. Please, describe it. I recommend adding the preliminary data for selecting the enzyme dosage as Supplementary material, so the authors can refer to it in the text without explaining much about it. The ones who are interested can check the supplementary data.
Lines 250-252: "For example, pretreated sugarcane bagasse by organosolv method, was 250 hydrolyzed at 10 FPU/g died substrate, while, other lignocellulosic materials are frequently hydrolyzed at 20-40 FPU/g [22]." Correct "died substrate". Also, I disagree that studies frequently use 20-40 FPU/g to hydrolyze biomass. Groups with experience in this field have been trying to reduce de cellulase dosage work under 10 FPU/g and several papers in the last decade have followed this trend.
The high glucan conversion obtained in relatively high dry matter (18-22%) could be related to the high amount of starch in samples coupled with the low content of lignin compared to studies where the glucan source was purely cellulose and had high lignin content. If the authors compare the yields obtained with studies using cane bagasse, wheat straw or woody biomass, they will see that the yields obtained in this range of dry matter (18-22%) are considerably high. However, no discussion is presented in this direction. Please, comment more on that.
Line 272 – Xylose hydrolysis should be "xylan hydrolysis". Xylose yield in Table 2 for 6.3% dry matter is 27.3%. I believe this value is incorrect. Please, check. The description of the material composition in percentage derived from NREL protocol would help analyze the results obtained in this section, as we are not provided with the glucan and xylan content. Some studies report that total xylan hydrolysis may also be complex if pectin is not degraded as well. I think the authors could revise the literature in this sense and make some hypotheses in this direction.
The hydrolysis profile in 25% DM differs from 22% DM. It was surprising to see a shift in initial hydrolysis with a slight increase in DM%. Could the authors discuss more regarding this phenomenon? Is some threshold achieved in this range of DM, such as viscosity or reduction in free water availability? It would be interesting to have more discussion regarding this.
Section 3.4. This section is very summarized and data is only presented in the simplified mass balance. I recommend expanding this section by giving more details and presenting the results in a figure/Table. Many studies have achieved excellent methane production yields from liquid fractions rich in organic acids after biohydrogen production. A recommend a comparison with liquids streams coming from such process.
Minor corrections:
Add error bars to the graphics, so we have a ideia of the data variation
Line 32 – Delete "ones"
Line 97-98 – Use italics for microorganism name
Line 106 – correlated
Line 133 - Use italics for microorganism name
Line 421 - – Use italics for microorganism name (check for this throughout the manuscript)
Author Response
Response to reviewers ‘comments
Reviewer #1
Comment: Line 51. It is better to define the concept of "source-separated municipal biowastes" clearly here, as this may not be obvious for people in countries where this practice is not common. Also, revise in the text the term "source-separated" that sometimes appears as "source separated". It would also be interesting to add some comments and data on the availability and logistics for using this material in a biorefinery in countries where source-separated municipal biowastes are collected.
OFHKW needs to be defined only once in the text. After that, make consistent use of the acronym created. (except in tables and Figures, where it is best to define it again)
Response: We are very grateful for this valuable comment. We added the following information in the revised manuscript in order to better define the concept of "source-separated municipal biowastes".
I know that source-separated municipal biowastes may consist of several sources, such as: kitchen household biowaste, restaurant leftovers, etc.). We clearly indicated that municipal biowaste used in this study represents organic fraction of household kitchen waste (OFHKW).
Consequently, the following information has been added to the revised manuscript:
„Source-separated municipal biowastes in the form of organic fraction of household kitchen waste (OFHKW)… “. We also defined OFHKW once and after that we used acronym created (besides tables and Figures), as requested. Please see the revised manuscript. We revise the text and corrected the term "source-separated" if necessary (your comment: sometimes appears as "source separated").
We agree that it would be interesting to present “data on the availability and logistics for using this material in a biorefinery in countries where source-separated municipal biowastes are collected”. However, the aim of this manuscript was to present integrated valuable compounds (succinic acid) and energy (biogas) production from organic fraction of household kitchen wastes (OFHKW), using for the first time simultaneous CO2 source (biogas after co-digestion processes and MgCO3). The study also includes integrated biogas production from by-products, after succinic fermentation and a simplified mass balance of OFHKW complex (1 Mg of waste treated).
The concept presented focuses on maximizing the products in a biorefinery approach. We also focused on enzyme selection, in order to obtain a high sugar concentration for succinic production. According to best of our knowledge, this is the first study evaluating usage of simultaneous CO2 source (biogas after co-digestion processes and MgCO3) for the production of succinic acid from municipal biowastes (OFHKW), with integrated biogas production from by-products after succinic fermentation.
However, we plan to develop the biorefinery concept in our future studies, including your valuable comment on „availability and logistics for using this material in a biorefinery in countries where source-separated municipal biowastes are collected”. We are very grateful for indicating this important aspect in your review. This would allow us to improve future manuscripts.
Comment: Section 2.1. How the sampling of OFHKW occurred? Did the authors get samples from different bins? How would changes in composition change the results obtained in the best conditions? Have you tested different samples of OFHKW to be sure that the process is robust enough, even with variations in the raw material composition? This is an essential factor to evaluate when proposing this type of raw material. Please, comment on that. It is better to describe the methods for characterizing the biomass here; otherwise, we see much-related information, but the characterization methods are listed only in the last topic. Also, please describe de glucan and xylan content (or the whole composition determined by the NREL protocol), as this is not shown in Table 1, and Table 2 presents results for glucose and xylose yields.
Section 3.1. As commented above, I think it is very important to comment on how this material composition changes when different sampling is done and how this impacts a biorefinery or facility based on the use of this material. How representative is the compositional data presented in this paper if one decides to use the biowaste from Bielsko-Biala, for example? Collecting samples at a different seasons would result in a very different composition? Please, include thoughts on that.
Response: Our aim was to obtain an average sample of feedstock used for experiments. We would like to indicate that our studies (presented in the manuscript) were aimed at establishing the optimal carbon dioxide dosage foe succinate production and to obtain valuable products in a biorefinery concept.
As regards the changes in composition of the feedstock used, there were no significant differences in glucan, xylan content (components used for succinic production). The argument raised in your review is very important, but we think that it is not a good idea to add characterization of each sample (month), taking into account the fact that succinic acid and biogas were produced from “an average sample” after mixing all waste portions. Such results (OFHKW characterization – each month) would not reflect other experimental results, presented in the whole manuscript.
Consequently, we added the following information into the revised manuscript:
“The OFHKW sample used in this study originated from biowaste bins locally provided in the municipality of Bielsko-Biala (south of Poland, about 170 000 inhabitants). The OFHKW was mainly consisted of food waste and it was collected at a municipality level once a week, from 20 households. Samples were collected between June and October (about 10 kg each, 20 OFHKW collections), from locations evenly distributed throughout the city of Bielsko-Biala. Once collected, the biowaste was mixed manually and a representative sample (about 1 kg from each collection) was stored in a refrigerator at - 4ºC. The sample used for enzymatic hydrolysis contained an even quantitative share of OFKKW from each collection (April-October, 20 individual OFHKW samples, based on weekly collections). The obtained sample was homogenized and autoclaved at 121 ºC for 1 h, before chemical characterization (Table 1) and enzymatic hydrolysis (Table 2)” (section 2.1., please see revised manuscript).
We added glucan and xylan content (determined by the NREL protocol), as this is not shown in Table 1, and Table 2.
Table 1 revised manuscript:
Carbohydrates %TS 42.7÷3.1
Cellulose %TS 29.2÷2.4
Starch %TS 7.10÷0.4
Hemicellulose %TS 6.40÷0.3
Comment: Section 2.2. The term "second-generation enzymes" may not be familiar to many readers, as this is a denomination in the context of enzyme development for cellulose hydrolysis on cellulosic ethanol production. It is also better to specify the activities of the enzymes and their sources in another way. Describing the FPU units in the middle of the description is confusing. From where Cellic enzymes were obtained? Were they purchased from Sigma or donated by Novozymes? Specify. Add the reference for Ghose method.
The enzyme dosage used (5% w/w) is unclear. As I understood, this 5% is in relation to the substrate dry matter content, i.e., if 100 g of substrate were to be used, then we need to add 5g of the enzyme cocktail. Is that so? I had to read the text more than once and combine it with the table information to reach this conclusion, so I guess it could be better described in the text. Also, the preparation of the enzyme cocktail could be better described. "100% cocktail dosage: 62.5% Cellic® 103 CTec2, 31%% β-Glucanase and 6.5% Cellic ® HTec2, expressed as a percentage of substrate 104 dry matter" is misleading. I guess the percentages are in relation to the total weight of the enzyme cocktail and not the substrate dry matter, right?
Section 3.2. Much information in here repeats what is already stated in the materials and methods. I recommend better describing the assays in the M&M, so everything is clear. Then, here, the authors can focus on discussing the results. Please, describe it. I recommend adding the preliminary data for selecting the enzyme dosage as Supplementary material, so the authors can refer to it in the text without explaining much about it. The ones who are interested can check the supplementary data.
Response: The above comments were combined as they mostly concern enzymes selection and rearranging “Materials and methods” section as regards enzymatic hydrolysis.
We agree with your comment as regards the term "second-generation enzymes". An appropriate information about enzymes activities and their sources were introduced. "Second-generation enzymes" has been removed from the whole manuscript in order to make the manuscript consistent.
We also put information about FPU units in the methods section, as requested. Additionally, the reference for Ghose method was added.
17.Siqueira, J.G.W., Teixeira, N.A., Vandenberghe, L.P.S. et al. Update and Revalidation of Ghose’s Cellulase Assay Methodology. Appl Biochem Biotechnol 191, 1271–1279 (2020). https://doi.org/10.1007/s12010-020-03291-0
We added information about the origin of the enzymes “Enzymatic cocktails were purchased from Sigma-Aldrich® (Poland)” (section 2.2, please see revised manuscript).
As a result, the following information was rewritten/included in the revised manuscript:
„In order to break down OFHKW complex into monomeric fermentable sugars for succinogenic bacteria; solid fraction of OFHKW was hydrolysed with application of β-Glucanase (from Trichoderma longibrachiatum – a mixture of enzymes with mainly β-1-3 and β-1-4-glucanase, xylanase, and cellulase activities, 40 FPU/g, G4423, Sigma-Aldrich) as well as Cellic® CTec2 (120 FPU/g) and Cellic® HTec2 (25 FPU/g) enzymatic cocktails” (section 2.2, please see revised manuscript).
We also agree with the comment on “The enzyme dosage used (5% w/w) is unclear…”. In fact, enzyme were added is in relation to the substrate dry matter content, as you kindly noticed.
We also agree that current information "100% cocktail dosage: 62.5% Cellic® 103 CTec2, 31%% β-Glucanase and 6.5% Cellic ® HTec2…” is misleading. This information has been rewritten as follows:
We would like to explain the way of enzyme/enzymatic mixtures selection, which was based on our preliminary experiments.
Firstly, the efficiency of commercially available cellulolytic enzyme preparations, using dosage of 2.5, 4.0, 5.0, 6.0, 7.5% w/w (expressed as percentage of substrate dry matter, substrate loading during preliminary experiments: 12.5%±1.2 and 25.0%±2,5; based on dry matter) was analyzed. Secondly, Cellic® CTec2 and β-glucanase, considered as the most efficient (glucan and xylan yield: > 75%), were added in a mixture, using various percentage of these two enzymatic cocktails (i.e. calculated in relation to the total weight of the enzyme cocktail: 5% and 6% w/w). The mixture of Cellic® CTec2 (two-thirds, % w/w) and β-glucanase (one-third, % w/w) with enzyme concentrations of 5% (total weight of all enzymatic cocktails), was considered as the most effective (glucan yields > 75%, xylan yields: > 60%). Finally, about 1/10 of the Cellic® CTec2 enzyme weight was replaced by Cellic® HTec2, in order to increase the xylan conversion into xylose. The obtained mixture of enzymatic cocktails (62.5% Cellic® 103 CTec2, 31% β-Glucanase and 6.5% Cellic ® HTec2) was used for OFHKW hydrolysis during studies (Table 2). Percentage of individual enzymatic components (Cellic® CTec2, β-Glucanase and Cellic ® HTec2) was calculated in relation to the total weight of the enzyme cocktail (i.e. 100% represents 5 g of the enzyme cocktails/100 g of OFHKW dry matter).
We also moved information about enzyme selection form „results and discussion section” and rearrange „Materials and Methods section”, according to the reviewer comments.
The following text was removed from section 3 „Results and discussion”: For reactions carried out with enzymes concentrations of up to about 4% (w/w) (preliminary results, data not shown), the amount of catalysts in the glucan and xylan hydrolysis is the only limiting factor i.e. reaction rates are dependent only upon the level of enzymes. For higher doses of Cellic CTec2/β-glucosidase, rates of both glucose as well as xylose release were not directly proportional to the amount of enzyme present in reaction mixture (the rates versus enzyme concentration response become nonlinear), most probably due to the saturation of substrate by enzymes [18]. Usage of enzyme concentrations above 5% (w/w) (>12.5 FPU/g-glucan) had no positive influence on the effectiveness of enzymatic hydrolysis. enzymatic dosage of 5% w/w (100% cocktail dosage: 62.5% Cellic® CTec2, 31%% β-Glucanase and 6.5% Cellic ® HTec2, expressed as enzymatic coctails weight) was considered as the most effective and applied for hydrolysis of solid fraction from OFHKW (Table 2).
Taking into account the above, the following text was introduced into the revised manuscript:
„Enzymatic dosage amounted to 5% w/w calculated in relation to the substrate dry matter content, i.e. 5 g of the enzyme cocktails/100 g of substrate (OFHKW) dry matter. Percentage of individual enzymatic components (Cellic® CTec2, β-Glucanase and Cellic ® HTec2) was calculated in relation to the total weight of the enzyme cocktail (100% enzymatic dosage represents, e.g. 5 g of the enzyme cocktails/100 g of OFHKW dry matter). The mixture of enzymatic cocktails used in this study (Table 2) contained: 62.5% Cellic® 103 CTec2, 31%% β-Glucanase and 6.5% Cellic ® HTec2, expressed as total weight of enzymatic cocktails. For reactions carried out with enzymes concentrations of up to about 4% (w/w) (preliminary results, data not shown), the amount of catalysts in the glucan and xylan hydrolysis is the only limiting factor i.e. reaction rates are dependent only upon the level of enzymes. For higher doses of Cellic CTec2/β-glucosidase, rates of both glucose as well as xylose release were not directly proportional to the amount of enzyme present in reaction mixture (the rates versus enzyme concentration response become nonlinear), most probably due to the saturation of substrate by enzymes [18]. Usage of enzyme concentrations above 5% (w/w) (>12.5 FPU/g-glucan) had no positive influence on the effectiveness of enzymatic hydrolysis. These enzymatic dosage correlated to between 0.3 and 1.25% of total sample volume (based results presented in Table 2) and cellulase activity of 12.5 FPU/g-glucan (10.5 FPU/g-carbohydrates or 4.5 FPU/g total biomass solids)” (section 2.2, please see revised manuscript).
We hope that the current information would be acceptable by the reviewers. We also added a reference to our previous manuscript, which describes a very similar way of enzyme selection. This publication includes supplementary material and can be used for calculations of enzyme selection.
- DÄ…bkowska, K.; Alvarado-Morales, M.; Kuglarz, M.; Angelidaki, I. Miscanthus straw as substrate for biosuccinic acid production: Focusing on pretreatment and downstream processing. Bioresour. Technol. 2019, 278, 2019, 82-91.
Comment: Table 2. What is the initial glucose and xylose content? I do not understand. Is that the theoretical maximum?
and
Line 272 – Xylose hydrolysis should be "xylan hydrolysis". Xylose yield in Table 2 for 6.3% dry matter is 27.3%. I believe this value is incorrect. Please, check. The description of the material composition in percentage derived from NREL protocol would help analyze the results obtained in this section, as we are not provided with the glucan and xylan content.
Response: We added glucan and xylan content (determined by the NREL protocol).
Table 1 revised manuscript:
Carbohydrates %TS 42.7÷3.1
Cellulose %TS 29.2÷2.4
Starch %TS 7.10÷0.4
Hemicellulose %TS 6.40÷0.3.
The glucose and xylose concentration (initial – before enzymatic hydrolysis) represent theoretical maximum.
We added the following description below Table 2:
b - theoretical (maximum) glucose and xylose concentration, after enzymatic hydrolysis and mixing with exponentially growing inoculum (5% v/v)
We also corrected the xylan yield in Table 2 (for 6.3% dry matter). We would like to apology for this mistake connected with xylan yield calculation. In fact, all concentrations of xylose are correct (Table 2). However, xylan yield should be 82.3±2.5%. This mistake was corrected in the revised manuscript. Total sugar concentration after enzymatic hydrolysis is not changed, the only mistake concerned xylan yield (please see the revised manuscript).
In the whole manuscript: Xylose hydrolysis was replaced with "xylan hydrolysis".
Comment: Section 2.5.1. Formula [1] – it is better to state "glucan" and not "cellulose" as glucose in hydrolysates may be derived from starch in the OFHKW and NREL protocol does not discriminate between starch and cellulose-derived glucan, unless previous hydrolysis with amylases is performed to determine the starch content separately. Did the authors determine the starch content in the residue? Considering that the biomass comes from kitchen waste, I expect a considerable amount of starch, which influences more efficient hydrolysis at a high dry matter than a pure-cellulosic source of glucose. Indeed, in lines 216-217 authors state, "Glucan content, which represents cellulose and starch, accounted for about 85% of all carbohydrates."
Response: We completely agree with your comment and Formula [1] was revised as indicated (please see revised manuscript, Formula 1).
We also confirm that "Glucan content, which represents cellulose and starch, accounted for about 85% of all carbohydrates.".
The starch content of OFHKW amounted to 7.1%±0.8, which can be considered as relatively low values, taking into account the feedstok used in our study. Starch content was included in the Table 1, which presents „Characterization of OFHKW (organic fraction of household kitchen waste) used in the current study (average values n = 3, ±standard deviation)”.
Comment: Lines 250-252: "For example, pretreated sugarcane bagasse by organosolv method, was 250 hydrolyzed at 10 FPU/g died substrate, while, other lignocellulosic materials are frequently hydrolyzed at 20-40 FPU/g [22]." Correct "died substrate". Also, I disagree that studies frequently use 20-40 FPU/g to hydrolyze biomass. Groups with experience in this field have been trying to reduce de cellulase dosage work under 10 FPU/g and several papers in the last decade have followed this trend.
Response: We completely agree with your comment as regards recent trend to minimize cellulase dosage.
The following text was rewritten:
“Therefore, applied enzyme dosages can be considered as relatively low, compared to other biomass types, hydrolysed at high biomass loadings. For example, pretreated sugarcane bagasse by organosolv method, was hydrolyzed at 10 FPU/g dried substrate. In general, due to the recent scientific progress in the field of enzymatic hydrolysis, the acceptable cellulase dosage used for biomass hydrolysis is around or even below 10 FPU/g DM [23]” (section 3.2, please see revised manuscript).
References added to the revised manuscript:
da Silva, A.S., Espinheira, R.P., Teixeira, R.S.S. et al. Constraints and advances in high-solids enzymatic hydrolysis of lignocellulosic biomass: a critical review. Biotechnol Biofuels 13, 58 (2020). https://doi.org/10.1186/s13068-020-01697-w
Comment: I am interested to know which of those enzymes contain amylases since the biomass used contains starch in its composition. Please, describe it. I recommend adding the preliminary data for selecting the enzyme dosage as Supplementary material, so the authors can refer to it in the text without explaining much about it. The ones who are interested can check the supplementary data.
And
The high glucan conversion obtained in relatively high dry matter (18-22%) could be related to the high amount of starch in samples coupled with the low content of lignin compared to studies where the glucan source was purely cellulose and had high lignin content. If the authors compare the yields obtained with studies using cane bagasse, wheat straw or woody biomass, they will see that the yields obtained in this range of dry matter (18-22%) are considerably high. However, no discussion is presented in this direction. Please, comment more on that.
Response: We analyzed starch content in feedstock and it contained 7.1±0.4 of dry matter, which is relatively low, compared to values frequently reported in the literature.
Values most frequently reported for kitchen biowaste vary from 4.7% (dry matter) to even 30% (dry matter) (Stylianou et al., 2020, López-Gómez et al., 2020).
References:
José Pablo López-Gómez, Cristina Pérez-Rivero, Joachim Venus. Valorisation of solid biowastes: The lactic acid alternative, Process Biochemistry, 99, 2020, p. 222-235.
Stylianou, E., Pateraki, C., Ladakis, D. et al. Evaluation of organic fractions of municipal solid waste as renewable feedstock for succinic acid production. Biotechnol Biofuels 13, 72 (2020). https://doi.org/10.1186/s13068-020-01708-w
Taking into account the initial content of starch in the feedstock, we did not apply dedicated enzymes for starch hydrolysis. As we know, it is not possible to obtain a valuable data on ingredients content of commercially available enzymatic cocktails. We are not able to show that the enzymes used contain any amylases as such data are not easily available (Sigma website, supporting materials).
Our study was focused on succinic process conditions (mostly carbon dioxide source and dosage) for effective glucose and xylose conversion into carboxylic acids). Taking into account the above, the aim of the enzymatic hydrolysis performer was to obtain the highest glucose and xylose concentration with biomass treated, at high dry matter loading.
During our studies, we improved xylan hydrolysis, by introducing additional enzymatic cocktails Cellic® HTec2 enzyme. Due to relatively low initial content of starch in our samples, we did not look into starch hydrolysis, by using dedicated enzymes. We are conscious that kitchen wastes can include a relatively high starch content, but it was not observed in our case. Compared to cellulose-derived glucan content, starch content seems to be relatively low (Table 1). That is why speculation in this regards was not included in the manuscript.
For exmple, a relatively low content of starch in kitchen waste was also observed in other studies, e.g. „Stylianou, E., Pateraki, C., Ladakis, D. et al. Evaluation of organic fractions of municipal solid waste as renewable feedstock for succinic acid production. Biotechnol Biofuels 13, 72 (2020). https://doi.org/10.1186/s13068-020-01708-w”. In this study, organic fraction of municipal solid waste contained from 4.71% (dry matter – spring/summer sample collection) to 5.31% (dry matter – autumn/winter sample collection).
Comment: The hydrolysis profile in 25% DM differs from 22% DM. It was surprising to see a shift in initial hydrolysis with a slight increase in DM%. Could the authors discuss more regarding this phenomenon? Is some threshold achieved in this range of DM, such as viscosity or reduction in free water availability? It would be interesting to have more discussion regarding this.
Response: Many thanks for your valuable comment. We completely agree with your arguments and it helped us to look into the results from additional side.
The following text was added into the revised manuscript:
“A significant decrease in glucan hydrolysis was observed after increasing dry matter loading from 22% to 25% (Table 2). Taking into account the fact that the difference between analyzed assays is relatively low, this phenomenon might have been caused by reduced viscosity or reduction in free water availability” (section 3.2, please see revised manuscript).
We also observed similar situation in our previous experiments, but it concerned lignocellulosic biomass (e.g. willow, hemp) and was mostly connected with mixing difficulties at high biomass loading (laboratory conditions). However, mixing difficulties were no observed in the current study.
Comment: Section 3.4. This section is very summarized and data is only presented in the simplified mass balance. I recommend expanding this section by giving more details and presenting the results in a figure/Table. Many studies have achieved excellent methane production yields from liquid fractions rich in organic acids after biohydrogen production. A recommend a comparison with liquids streams coming from such process.
Response: We would like to explain that presented mass balance (biorefinery concept as a summary) is a simplified one and it is based on the residues after succinic acid production. It only contains biogas production from residues after succinic fermentation at optimized conditions (carbon dioxide source, succinic fermentation with 25% of theoretical MgCO3 dosage, simultaneous addition of biogas and MgCO3: 20-21 g/L). It is not possible to present biogas production from residues after all succinic assays (Table 3) within additional tables.
In this conditions, we obtained “methane yields fluctuated between 0.200 and 0.220 Nm3 CH4/kg VSadded (208±11Nm3 CH4/kg VSadded ) (Fig. 3)”. It should be taken into account that biogas was produced from post-hydrolysis residues and liquid fractions OFHKW. Such substrates are not reach in organic acids (liquid fraction Table 1: VFA 0.55 g/L). In fact, organic acids are present in succinic broths (succinic acid, acetic and formic acids) and this fraction should be directed to downstream processes and purification. This fraction was not used as substrate for biogas production. In our opinion, it is not a good idea to compare methane yields with liquid fractions rich in organic acids, e.g. after biohydrogen production. We hope that this explanation would be accepted, but we agree that the presented mass balance/biorefinery concept is a simplified one. However, the aim of this part of the paper was to show the concept aimed at maximizing the valuable products as well as carbon dioxide, which is used during succinic fermentation, by Actinobacillus succinogenes.
Minor corrections:
Comment: Add error bars to the graphics, so we have a idea of the data variation.
Response: Error bars are included. Please see Fig. 1 and 2.
Comment: Line 32 – Delete "ones"
Response: Corrected.
Comment: Line 97-98 – Use italics for microorganism name
Response: Corrected.
Comment: Line 106 – correlated
Response: Corrected.
Comment: Line 133 - Use italics for microorganism name
Response: Corrected.
We are very grateful for all your valuable comments. We tried to introduce them into the revised manuscript, to the larger possible extend. Some comments are very difficult to apply as the succinic acid production and biorefinery concept is based on an average/representative sample. We are sure that your comments will help us to prepare better manuscripts in the future. We hope that the revised manuscript is significantly improved, especially methods description.

Reviewer 2 Report
Well written paper with important and valuable results. Minor comments:
Line 84: How were the liquid and solid fractions obtained from the raw OFHKW sample? And a minor grammar error: Organic fraction of household kitchen waste (OFHKW) was…
Line 96: ‘;’ Should be removed or replaced by ‘,’.
Table 2: It contains experimental results and could be transferred to part 3. Results and discussion. The same for Table 3.
Lines 148-149: How were co-digestion experiments carried out?
Line 227: ‘;’ Should be removed or replaced by ‘,’.
Line 272: For 20% wet substrate loading, xylose yield was much lower than 61% according to Table 2.
Figure 3: The sum of solid and liquid fractions slightly exceeds that of the initial OFHKW weight.
Author Response
Response to reviewers´comments
Reviewer #2
Comment: Line 84: How were the liquid and solid fractions obtained from the raw OFHKW sample? And a minor grammar error: Organic fraction of household kitchen waste (OFHKW) was…
Response: Liquid and solid fractions were obtained from the raw OFHKW sample, via a low speed centrifuge (Benchtop, CFG-5BL, 2000 rpm).
We added the following information into the revised manuscript:
“In order to get homogenized composition, the OFHKW was first freeze-dried and then blended in a food processor to powder form and was stored at 4 °C before usage. OFHKW was separated into solid (85% of total sample weight) and liquid fractions (15% of total sample weight) (Table 1). ), using a low speed centrifuge (Benchtop, CFG-5BL, 2000 rpm)”.
Indicated grammar error has been corrected (please see revised manuscript).
Comment: Line 96: ‘;’ Should be removed or replaced by. Line 227: ‘;’ Should be removed or replaced by.
Response: Corrected.
Comment: Lines 148-149: How were co-digestion experiments carried out?
Response: We added co-digestion conditions into the revised manuscript.
Consequently, the following text was added into the revised manuscript:
“The biogas originated from our previous sewage sludge and kitchen biowaste co-digestion experiments, conducted at lab scale in a thermophilic conditions (52 ºC). The biogas sample was collected during fermentation of kitchen biowaste (60% dry matter of the feedstock) and sewage sludge (40% dry matter of the feedstock), at optimized conditions (HRT = 25 days)” (section 2.3).
Comment: Table 2: It contains experimental results and could be transferred to part 3. Results and discussion. The same for Table 3.
Response: We agree with this comment. Table 2 and 3 were transferred to part “3. Results and discussion (please see the revised manuscript).
Comment: Line 272: For 20% wet substrate loading, xylose yield was much lower than 61% according to Table 2.
Response: We would like to apology for this mistake connected with xylan yield calculation. In fact, all concentrations of xylose are correct (Table 2). However, xylan yield should be 82.3±2.5%. This mistake was corrected in the revised manuscript. Total sugar concentration after enzymatic hydrolysis is not changed, the only mistake concerned xylan yield.
Comment: Figure 3: The sum of solid and liquid fractions slightly exceeds that of the initial OFHKW weight.
Response: Values of liquid fractions (dry matter) and solid fractions (dry matter) represent average values from 3 measurements. I agree with your comment that „the sum of solid and liquid fractions slightly exceeds that of the initial OFHKW weight”. For example, the difference as regards dry matter is 0.2% (compared to initial OFHKW weight). Similar difference has been observed in case of volatile solids (VS) content. In our opinion, taking into account the nature of the substrate used, the differences noticed are insignificant from the technological point of view. All results presented in simplified mass balance (biorefinery concept) are based on laboratory analyses.
